# A Super-Clustering Approach for Fully Automated Single Particle Picking in Cryo-EM

**DOI:** 10.3390/genes10090666

**Published:** 2019-08-30

**Authors:** Adil Al-Azzawi, Anes Ouadou, John J. Tanner, Jianlin Cheng

**Affiliations:** 1Electrical Engineering and Computer Science Department, University of Missouri, Columbia, MO 65211, USA; 2Departments of Biochemistry and Chemistry, University of Missouri, Columbia, MO 65211, USA; 3Informatics Institute, University of Missouri, Columbia, MO 65211, USA

**Keywords:** super-clustering, intensity based clustering (IBC), micrograph, cryo-EM, singe particle pickling, protein structure determination, k-means, fuzzy c-means (FCM)

## Abstract

Structure determination of proteins and macromolecular complexes by single-particle cryo-electron microscopy (cryo-EM) is poised to revolutionize structural biology. An early challenging step in the cryo-EM pipeline is the detection and selection of particles from two-dimensional micrographs (particle picking). Most existing particle-picking methods require human intervention to deal with complex (irregular) particle shapes and extremely low signal-to-noise ratio (SNR) in cryo-EM images. Here, we design a fully automated super-clustering approach for single particle picking (SuperCryoEMPicker) in cryo-EM micrographs, which focuses on identifying, detecting, and picking particles of the complex and irregular shapes in micrographs with extremely low signal-to-noise ratio (SNR). Our method first applies advanced image processing procedures to improve the quality of the cryo-EM images. The binary mask image-highlighting protein particles are then generated from each individual cryo-EM image using the super-clustering (SP) method, which improves upon base clustering methods (i.e., k-means, fuzzy c-means (FCM), and intensity-based cluster (IBC) algorithm) via a super-pixel algorithm. SuperCryoEMPicker is tested and evaluated on micrographs of β-galactosidase and 80S ribosomes, which are examples of cryo-EM data exhibiting complex and irregular particle shapes. The results show that the super-particle clustering method provides a more robust detection of particles than the base clustering methods, such as k-means, FCM, and IBC. SuperCryoEMPicker automatically and effectively identifies very complex particles from cryo-EM images of extremely low SNR. As a fully automated particle detection method, it has the potential to relieve researchers from laborious, manual particle-labeling work and therefore is a useful tool for cryo-EM protein structure determination.

## 1. Introduction

For decades, X-ray crystallography and nuclear magnetic resonance (NMR) spectroscopy have been the principal technologies of high resolution structural biology, accounting for over 95% of the current holdings of the Protein Data Bank. However, in the past few years, major technological advances have fueled a “resolution revolution” in cryo-electron microscopy (cryo-EM) [1,2,3], and cryo-EM has emerged as a leading structural biology technology capable of determining protein structures to resolutions rivaling X-ray crystallography [4,5,6,7,8,9].

Identification of particles in micrographs (particle picking) is a critical step in structure determination by cryo-EM. The micrographs result from passing an electron beam through a thin vitrified sample to create 2D image projections of the particle under study [10]. Ultimately, the 3D shape (density map) of the protein is reconstructed from the 2D images. The 2D cryo-EM images contain randomly arranged particles along with non-particles—bits of frost, deformed particles, protein aggregates, and so on. These images have high background noise and low contrast due to a limited electron dose used in imaging. A large number of single-particle images need to be picked from cryo-EM micrographs to perform a reliable 3D reconstruction of the underlying protein structure. Particle picking thus represents an early bottleneck in the practice of cryo-EM structure determination.

Particle picking methods can be basically divided into three categories: Generative methods [11,12], discriminative classification [13,14,15,16], including the recent deep learning approaches [17,18], and unsupervised learning (clustering) methods [19]. Typically, the generative method employs a template-matching technique, which measures the similarity to a reference to identify particle candidates from micrographs. This technique requires initial high-quality particle templates which are manually selected by an expert. The discriminative classification technique requires preparing an initial set of manually labeled reference particles as the training dataset to train a classifier (e.g., a deep convolutional neural network) to detect particles. Therefore, generative and discriminative methods are not fully automated.

A typical generative method employs a template-matching technique with a cross-correlation similarity measure to accomplish particle selection. Template-based matching methods are very sensitive to noise and result in a substitutional fraction of false positives since the template-based matching methods rely on local cross-correlation, which result from a false correlation peak [20]. Thus, some initial “good references” are selected in advance to ensure that the manually selected examples have less noise compared with the other particles in the same (2D) micrographs. The discriminative methods first train a classifier based on a labeled dataset of positive and negative examples and then apply this trained classifier to detect and recognize particle images from micrographs. Also, some “good examples” are selected in advance, which avoid the low-contrast particle examples from the micrographs. In most cases, the “bad particle” examples include the local aggregates, overlapped particles, background noise fluctuations, carbon-rich areas, and ice contamination. Thus, after initializing the classifier, an additional step called “manual versification and selection” is required to sort out the “good examples” and isolate them from the “bad ones” [21]. In contrast, the unsupervised approaches distinguish the images of particle-like objects from background noise in micrographs via an unsupervised learning manner (i.e., without any labeled training data). Therefore, the unsupervised approaches are often combined with the template-matching or classification-based approaches to achieve decent picking results [22].

To aid the streamlining of particle picking, we propose a super-fully automated approach (SuperCryoEMPicker) for picking single particles of complex shapes in cryo-EM images, leveraging the new super-clustering technique. This method improves the base clustering algorithms (e.g., k-means) using the super-pixel algorithm (simple linear iterative clustering (SLIC)) [23]. Specifically, the super-clustering algorithm applies a base clustering algorithm such as k-means [24], fuzzy c-means (FCM) [25], or intensity-based clustering (IBC) [26] to generate a super-pixel map, which is then used for fully automated particle picking in cryo-EM images without human intervention. We demonstrate that our fully automated super-clustering approach can accurately detect and select a sufficient number of complex particles that are comparable to those picked manually. Therefore, it can significantly reduce time and labor spent on particle picking and relieve a bottleneck in the cryo-EM structure determination pipeline.

## 2. Methods

Some cryo-EM datasets feature particles with relatively simple shapes, such as circles or squares; our recently reported algorithm, AutoCryoPicker [26], was designed to handle these cases. Herein, we address the more general (and challenging) case of particles with arbitrary shapes [14,27]. In this case, detecting and picking the irregular or complex particle shapes in very low signal-to-noise ratio (SNR) cryo-EM images faces two main challenges. First, particles appear as non-structural object shapes, which makes template-matching algorithms unable to distinguish between the objects and the background. Second, the particles in the very low SNR cryo-EM images have almost the same intensity level as the background. To address this problem, we propose a super-fully automated approach (SuperCryoEMPicker) for picking single particles of complex shapes in cryo-EM images, leveraging the new super-clustering technique.

The super-clustering approach was designed especially for picking particles of irregular and complex shapes, as shown for the 80S ribosome [28] in Figure 1a,b. The framework of the super-clustering approach is shown in Figure 2. It was divided into three main stages—pre-processing, particle clustering, and particle picking. In the first stage (pre-processing), the 3D grid (array) of voxels MRC file (Medical Research Council) was converted to the PNG (Portable Network Graphics) image file format using EMAN2 (Executable Image Suite for Electron Microscopy) [29] in order to apply various image pre-processing techniques. Then, the same advanced pre-processing steps were used to improve the quality of the cryo-EM images. In the second stage, the binary mask of the cryo-EM image was generated. Two kinds of clustering methods were implemented in this stage. The first kind was the base clustering method, which included k-means [24], FCM [25], or intensity-based clustering (IBC) [26]. The second kind was the super-clustering (SP) approach (super-pixel-based simple linear iterative clustering (SLIC) [23]), which was implemented to improve the three base clustering algorithms (k-means, FCM, and IBC), leading to three super-clustering algorithms (SP-k-means, SP-FCM, and SP-IBC). In the third stage, based on the generated binary mask, a final set of particles was selected and picked from clustered particle candidates after some post-processing steps, such as binary mask cleaning and particle property measurement.

### 2.1. Stage 1: Pre-Processing

In this stage, the EMAN2 software [29] was used to adjust the global intensity of the cryo-EM micrographs and convert them from the MRC file format to the PNG image format in order to apply standard image-processing tools to them. Figure 3 shows some samples where different scaling factors were used with EMAN2 [29] to adjust the intensity of the cryo-EM images.

Figure 3b shows the same zoomed-in particle image after using scale factor 5 and its histogram, respectively, which had better contrast than the original image, shown in Figure 3a. However, the quality of images in Figure 3h which were adjusted with scaling factors 0.1 and 0.5, were lower than the original.

In the second step, different image pre-processing procedures (image resolution, global intensity adjustment, global contrast enhancement-based histogram equalization, noise suppressing using the Wiener filter, local particle contrast enhancement with adaptive histogram equalization, and edges enhancement using guided image filtering) were applied to improve the quality of the cryo-EM images as in AutoCryoPicker [26]. The results of the pre-processing procedures for the ribosome [28] and beta-galactosidase [30] images are shown in Figure 4.

### 2.2. Stage 2: Particle Clustering

In order to pick each possible particle in the cryo-EM image, a binary mask that clusters particles was needed. Both the base clustering algorithms (k-means [24], FCM [25], and IBC [26]) and the super-clustering algorithm built on top of the base algorithms (SP-K-means, SP-FCM, and SP-IBC) via pixel posterization using simple linear iterative clustering (SLIC) [23] were used to create the binary mask according to the following steps.

#### 2.2.1. Clustering with Base Clustering Algorithms

We applied three base clustering algorithms, including k-means, FCM, and IBC, to cluster particles. The number of clusters was chosen based on the predefined cluster numbers that the ICB clustering algorithm defined [26]. The initial number of clusters in the ICB algorithm was based on two factors—the adjusted intensity range and the interval size. The adjusted intensity range was automatically computed from the pre-processed cryo-EM image based on the lower and the upper bounds of the intensity level, while interval size was computed based on the ratio between the difference of the maximum and minimum intensity level and the adjusted intensity range. For instance, the initial number of clusters was *K* = 4, and if the adjusted intensity range was from 0.2 to 0.8 and the interval size was 0.15, there were 4 initial cluster levels, where the intensity level [0.2–0.35] is assigned to Cluster 1, [0.35–0.5] to Cluster 2, [0.5–0.65] to Cluster 3, and 316 [0.65–0.8] to Cluster 4. Figure 5 illustrates the results of the three base clustering algorithms. More details about applying the three algorithms to particle clustering can be found in [26].

#### 2.2.2. Super-Particle Clustering

We designed a super-clustering approach to further improve cryo-EM binary mask image generation based on pixel posterization using simple linear iterative clustering (SLIC) [23]. In this approach, an intermedia image map (super pixel over segmentation image) was generated and used as the input for the three base clustering algorithms (k-means, FCM, and IBC) to perform clustering, leading to three super-clustering methods, SP-k-means, DP-FCM, and SP-IBC, for fully automated single particle picking in cryo-EM.

The SLIC super-pixel method [23] combines the two kinds of distances between pixels i and j (Equation (1) for intensity distance and Equation (2) for spatial distance) into a single distance in Equation (3) [23]:(1)$$d_{c}=\sqrt{{\left(l_{j}-l_{i}\right)}^{2}},$$
(2)$$d_{s}=\sqrt{{\left(x_{j}-x_{i}\right)}^{2}+{\left(y_{j}-y_{i}\right)}^{2}},$$
where l is the intensity level and x and y are spatial pixel information.
(3)$$D^{\prime }=\sqrt{{\left(\frac{d_{c}}{N_{c}}\right)}^{2}+{\left(\frac{d_{s}}{N_{s}}\right)}^{2}},$$
where $N_{c}$ and $N_{s}$ are the maximum distance within a cluster. Basically, SLIC normalizes the distances of the intensity value and the spatial information by their respective maximum values. The maximum spatial distance is calculated based on the expected spatial distance within a given cluster that corresponds to the sampling grid interval (S). To produce the roughly equal-sized super-pixels, the interval grid S is calculated as is shown in Equation (4) [23]:(4)$$N_{s}=S=\sqrt{\frac{N}{k}},$$
where k is the desired number of the approximated equal-sized super-pixels and N is the lowest gradient position (3 × 3) neighborhood.

To avoid centering a super-pixel in an edge, the centers of the grid interval (S) are moved to the seed location corresponding to the lowest gradient position (3 × 3) neighborhood. To reduce the distance computations, SLIC only computes the distance from each pixel to each cluster center within the 2S×2S region. Since the intensity level can vary significantly from image to image and from cluster to cluster, the calculation of the maximum intensity distance $N_{c}$ is straightforward. $N_{c}$ is fixed as a constant m so that the weighted distance measure is calculated as Equation (5) [23].
(5)$$D^{\prime }=\sqrt{{\left(\frac{d_{c}}{m}\right)}^{2}+{\left(\frac{d_{s}}{S}\right)}^{2}},$$
where m is a fixed constant number and S is the grid interval. Multiplying Equation (5) by $\sqrt{m^{2}}$ leads to a simplified distance measure in Equation (6):(6)$$D^{\prime }=\sqrt{d_{c}{}^{2}+{\left(\frac{d_{s}}{S}\right)}^{2}m^{2}}.$$

Also, m allows us to weight the relative importance of the spatial information and the intensity similarity. When m is large, there is a small area-to-perimeter ratio (more impact); otherwise, there is an area close to the image boundary, which is less regular. Figure 6 shows examples of different intermediate cryo-EM images of super-pixel clustering for ribosomes [28] and beta-galactosidase [30]. We compared the intermediate cryo-EM images generated from the original cryo-EM images without pre-processing and with the ones from the pre-processed images (Figure 6).

The intermediate images generated from original images were worse than the originals, but the intermediate cryo-EM images generated from the pre-processed images were better than the pre-processed images. The intermediate images generated from the pre-processed images were used by the three base clustering algorithms for super-clustering, respectively. The super-pixel k-means clustering (SP-k-means) had three significant steps: Generating the 2D super pixels from the intermediate images, using k-means to cluster super-pixels (i.e., binary mask generation), and selecting the particle cluster from the clean image mask without small objects.

The SP-k-means automatically selected the proper cluster number for particles after the non-zero element (clustered particles) was extracted from each cluster image, as is shown in Figure 7. The SP-k-means did not require extra human intervention to select the most appropriate cluster like the original k-means [24] did. SP-k-means is a fully automated clustering algorithm based on the assumption that each group of white pixels (non-zero elements) represents one particular single particle in different cluster image and the black pixels represent the cryo-EM background (see Figure 7b). SP-k-means labeled each group of pixels (particles) in each cluster image (binary) and numbered each single particle (see Figure 7c). Based on the total number of particles in each cluster image (see Figure 7d–g), the SP-k-means selected the optimal cluster that had the smallest number of labels, which represented the target cluster that had the correct particle number and position in the original image (see Figure 7f).


**Algorithm 1: SP-k-Means Clustering Algorithm**
1. **Input:** Pre-processed cryo-EM image $I_{p}$2. **Return:** Super-clustered image $I_{sc}$3. Set number of clusters, K4. Generate the 2D super-pixel from the super-clustered image5. **begin**/* SLIC */6.  Initialize $C_{k}={\left[l_{k},x_{k},y_{k}\right]}^{T}$/* the cluster centers */7.  Move the cluster center cluster centers to the lowest gradient position in a 3 × 3 neighborhood.8.  Set label $l\left(i\right)=-1$ for each pixel i.9.  Set distance $d\left(i\right)=\infty$ for each pixel i.10.  **repeat**11.   **for**
k=1 to K
**do**12.    **for** each pixel i in 2S×2S region around $C_{k}$
**do**13.     Compute distance D between $C_{k}$ and i.14.      **if**
$D<d\left(i\right)$
**then**15.       set $d\left(i\right)=D$.16.       set $l\left(i\right)=k$.17.      **end if**18.    **end for**19.   **end for**20.   Compute new cluster center $\theta_{k}$.21.   Compute residual error E.22.  **until**
E≤threshold23.  generate binary mask24.  end/* SLIC */25.  **repeat**26.   **for** n = 1 to N
**do**27.    Determine the closest representative, $\theta_{k}$, for $x_{n}$28.    Set label for data point n to k29.   **end for**30.   **for** k = 1 to *K*
**do**31.    Update cluster representative $\theta_{k}$ to the mean with cluster label k32.                  $\theta_{k}=\frac{\sum_{n=1}^{N}u_{nk}x_{n}}{\sum_{n=1}^{N}u_{nk}}$33.   **end for**34.  **until** change in cluster centers are small35.   **for** k = 1 to to *K* do/* foe each clustered image */36.   $I_{\mathrm{sc}}\leftarrow \mathrm{Min}\left(\mathrm{Nonzero}\left(I_{k}\right)\right)$/* extract the total number of the non-zero element in each cluster and select the cluster that has the minimum total number of the non-zero element */37.   **end for**

Similarly, SP-FCM and SP-IBC are described in Algorithms 2 and 3, respectively.


**Algorithm 2: SP-FCM Clustering Algorithm**
1. **Input:** Pre-processed cryo-EM image $I_{p}$2. **Return:** Super-clustered image $I_{sc}$3. Set number of clusters, k4. Generate the 2D super-pixel over segmentation image5.  **begin**/*SLIC*/6.  Initialize the cluster centers $C_{k}={\left[l_{k},x_{k},y_{k}\right]}^{T}$7.  Move the cluster center cluster centers to the lowest gradient position in a 3 × 3 neighborhood.8.  Set label $l\left(i\right)=-1$ for each pixel i.9.  Set distance $d\left(i\right)=\infty$ for each pixel i.10.  **repeat**11.   **for**
k=1 to K
**do**12.    **for** each pixel i in 2S×2S region around $C_{k}$
**do**13.     Compute distance D between $C_{k}$ and i.14.     **if**
$D<d\left(i\right)$
**then**15.      Set $d\left(i\right)=D$.16.      Set $l\left(i\right)=k$.17.     **end if**18.    **end for**19.   **end for**20.   Compute new cluster center $\theta_{k}$.21.   Compute residual error E.22.  **until**
E≤threshold23. generate binary mask24. end/* SLIC */25. **repeat**26.   **for** n = 1 to N do27.    Update membership $u_{\mathrm{nk}}$ by taking sum of distance ratios of cluster k and all clusters.28.                 $u_{nk}=\overset{K}{\underset{i=1}{\sum }}{\left(\frac{d\left(x_{n},\theta_{k}\right)}{d\left(x_{n},\theta_{i}\right)}\right)}^{-\frac{1}{m-1}}$29.       **end for**30. **until** change in cluster centers are small31. **for** k = 1 to K
**do**/* foe each clustered image */32.    $I_{\mathrm{sc}}\leftarrow \mathrm{Min}\left(\mathrm{Nonzero}\left(I_{k}\right)\right)$/* extract the total number of the non-zero element in each cluster and select the minimum one as a final selected clustered image */33. **end for**


**Algorithm 3: SP-IBC Clustering Algorithm**
1. **Input:** Pre-processed cryo-EM image $I_{p}$2. **Return:** Super-clustered image $I_{sc}$3. Set number of clusters, k4. Generate the 2D super-pixel over segmentation image5.  **begin**/* SLIC */6.  Initialize the cluster centers $C_{k}={\left[l_{k},x_{k},y_{k}\right]}^{T}$7.  Move the cluster center cluster centers to the lowest gradient position in a 3 × 3 neighborhood.8.  Set label $l\left(i\right)=-1$ for each pixel i.9.  Set distance $d\left(i\right)=\infty$ for each pixel i.10.  **repeat**11.   **for**
k=1 to K
**do**12.    **for** each pixel i in 2S×2S region around $C_{k}$
**do**13.     Compute distance D between $C_{k}$ and i.14.     **if**
$D<d\left(i\right)$
**then**15.      Set $d\left(i\right)=D$.16.      Set $l\left(i\right)=k$.17.    **end if**18.    **end for**19.   **end for**20.   Compute new cluster center $\theta_{k}$.21.   Compute residual error E.22.  **until**
E≤threshold23.  Generate binary mask24. **end**/* SLIC */25. Transform the intermedia 2D image map $I_{m}$ into 1D $I_{v}$ which has the intensity values of all the pixels.26. L← height × width where L is the total number of pixels in the $I_{p}$27. $V_{\mathrm{Max}}$← $\mathrm{Max}\left[I_{v}\right]$/* maximum values of intensity in the image */28. $V_{\mathrm{Min}}$← $\mathrm{Min}\left[I_{v}\right]$/* minimum values of intensity in the image */29. K← 4/* set the number of initial cluster based on the interval size = 0.15 */30. **for**
i = 1 to K
**do**31.  ${\mathrm{Int}}_{s}$← $\left(\frac{V_{\mathrm{Max}}-V_{\mathrm{Min}}}{K\times 0.15}\right)\times 0.15/$* set the interval size as the vector of intensity range (max min) divided by the cluster number K */32. **end for**33. **for**
i = 1 to K
**do**34.  $\theta_{k}\leftarrow {\mathrm{Int}}_{s}\left[i\right]$ /* initialize the cluster center based on the interval size */35. **end for**36. **repeat**37.    **for**
i = 1 to K
**do**38.    **for**
j = 1 to L do39.     $\mathrm{Cluster}\left[\mathrm{ind}\right]\;\leftarrow \min(\mathrm{abs}\left(\theta_{k}\;\left[i\right]-{\;I}_{v}\;\left[j\right]\right)$/* assign $x_{n}$ the cluster k whose center ($\theta_{k})$ is closest to $x_{n}$ according to the absolute intensity difference between the two */40.   **end for**41.  **end for**42.  **for**
n = 1 to K
**do**43.   $\theta_{k}\left[n\right]\leftarrow \sum_{n=1}^{k}\frac{\mathrm{Cluster}\left[k\right]}{\mathrm{count}\left(\mathrm{ones}\right)}$/* update the mean $\theta_{k}$ of each cluster by calculating the average intensity values of the pixels assigned to the cluster */.44.  **end for**45. **until** there is no change in cluster centers.

Figure 8 shows some examples of the super-clustering results for the 80S ribosome [28] and beta-galactosidase [30], comparing them with the base clustering algorithms. Figure 8 shows that the super-clustering results were better than those of the base algorithms. The three super-clustering methods were fully automated, generated cleaner image masks, and ran faster than their base clustering algorithms. For instance, the running time of clustering one image using k-means was 46.07 s, versus 117.06 s for SP-k-means.

### 2.3. Stage 3: Particle Picking

The particle picking stage had two main steps. The first step was the binary mask image cleaning and the second step was the particle detection and picking. The image masks generated by the super-clustering methods (e.g., Figure 8f,h,j,l,n,p) were clean and did not require a post-processing stage to clean them. However, the binary mask image cleaning step was required to remove some small and noisy objects from the image mask generated by the base clustering algorithms (e.g., Figure 8e,g,i,k,m,o).

#### 2.3.1. Binary Mask Cleaning

The regular binary mask for each cryo-EM cluster image generated by a base clustering method (k-means, FCM, and IBC) was cleaned through the removal of the small and non-connected objects. The image cleaning algorithm is shown in Algorithm 4.


**Algorithm 4: Image Cleaning**
1. **Input:**
$I_{c}$/* cluster cryo-EM image */2. **Return:**
$I_{cc}$/* cleaned cluster image */3. $I_{c1}\leftarrow \mathrm{imopen}\left(I_{c}\right)$/* apply image opening on the cluster image to enlarge small blobs */4. $L\leftarrow \mathrm{bwlabel}\left(I_{c1}\right)$/* label each object in the cluster image which returns a label matrix L that contains 8-connected object in the cluster */5. **for**
i=1 to L
**do**/* for each object in the clustered image */6.  $I_{\mathrm{object}}\leftarrow \mathrm{state}\left(L\left(k\right)\right)$/* get each particle where k is the total number of objects in the cluster cryo-EM */7.  $I_{\mathrm{object}}\leftarrow \mathrm{remove}\left(\mathrm{state}\left(L\left(k\right)\right)\;\right)$/* remove all the connected components that are smaller than the p pixels */8.  $\mathrm{centroid}=\mathrm{stats}\left(k\right).\mathrm{Centroid}$/* mark and get the actual index numbers and the centroid of each object */9. **end for**10. ${\mathrm{obj}}_{\mathrm{number}}\leftarrow \mathrm{is}\;\mathrm{member}\left(I_{\mathrm{object}}\right)$/* extract the number of object (particles) */11. L←bwlabel/* label each object (particle) */12. **for**
i = 1 to L
**do**/* for each object (particles) */13.  Do size filtering and roundness filtering14.  $p\;\leftarrow \left[\mathrm{props}.\mathrm{label}\right]$/* extract the 8-connectedlabeld object */15.         keeperObjects←props.label> p/* remove each object that less than 8-connected component in the binary image */16.  $I_{c2}\leftarrow \mathrm{keeperObjects}\;$/* get actual index numbers instead of a logical vector */17.  $I_{\mathrm{cc}}\leftarrow \mathrm{bwareaopen}\left(I_{c2}\right)$/* produce new binary image with only the small, or non-connected object */18. **end for**19. Generate and getting a new binary image

Figure 9 shows examples of the image cleaning and non-connected object removal on the image masks generated by the three base clustering algorithms. Figure 9 shows that image cleaning separated the particles from the background noise in the image masks generated by the three base clustering methods well.

Figure 10 shows the whole particle images of the ribosome [28] and beta-galactosidase [30] before and after image cleaning for k-means [24], FCM [25], and IBC [26] clustering, respectively. For instance, Figure 10a,b shows the whole particle ribosome [28] image before and after image cleaning for k-means, respectively. Figure 10c,d shows the whole particle image of beta-galactosidase before and after image cleaning for k-means respectively. In both cases, we noticed that when the non-particles (small objects) and noise in the clustering image were cleaned and removed from the background, the objects (particles) became clearer to detect and pick.

#### 2.3.2. Single Particle Detection and Picking

Since the shapes of the protein particles were complex or irregular, particle detection and picking step was applied on clean image masks to detect each single particle. The particle detection and picking algorithm is described in Algorithm 5.


**Algorithm 5: Single Particle Detection and Picking**
1. **Input:**
$I_{cs}$ /* cleaned cluster image with square shapes only */2. **Return:**
$I_{cps}$ /* cleaned cluster image with perfect square shapes */3. $L\leftarrow \mathrm{bwlabel}\left(I_{c1}\right)$ /* label each object in the cluster image */4. **for** i = 1 to L
**do** /* for each object in the clustered image */5.  $\mathrm{Stats}\;\leftarrow \;\mathrm{regionprops}\left(I_{\mathrm{cs}}\right)$/* measure properties of particle region */6.   Areas ← [props.Area /* compute all the shape measurements and the pixel value measurements as well */7. **end for**8. **for**
i = 1 to $size\left(keeperObjects\right)$
**do**/* for each particle object */9.  $\left[x,y\right]\leftarrow centroid\left(keeperObjects\right)$/* extract the centroid is the horizontal coordinate (or x-coordinate) and vertical coordinate (or y-coordinate) */10.  Draw all bounding box for a discontinuous region11. **end for**

The detection algorithm returned a bounding box drawn around each particle detected in the image. Figure 11 shows the detection results of two different cryo-EM images of the ribosome [28] and beta-galactosidase [30]. The results of the super-clustering methods were significantly better than the base clustering methods for both the ribosome [28] and the beta-galactosidase [30] images.

## 3. Results and Discussion

### 3.1. Datasets

Images from two datasets (80S ribosome [28] and beta-galactosidase [30]) were used to evaluate the fully automated particle picking using the base and super-clustering methods. The two datasets were download form the Electron Microscopy Public Image Archive (EMPIAR-10028 and EMPIAR-10017). The ribosome dataset (EMPIAR-10028) [28] was in a multi-frame MRC image format (32 Bit Float). The size of each micrograph was 4096 by 4096 pixels and the dataset consisted of 1081 micrographs, each with 16 frames per image. The beta-galactosidase dataset (EMPIAR-10017) was in the single-frame MRC image format (32 Bit Float). It was a public dataset available in the Electron Microscopy Public Image Archive (EMPIAR) as EMPIAR-10017. The size of each micrograph was 4096 by 4096 pixels, with 84 micrographs in total. The micrographs were the average, without any realignment, of 24 raw movie frames (accumulating 24 electron per squared Angstrom in a 1.5 s exposure). For the experimental results, we randomly selected 80 micrographs from the ribosome dataset [28] and 80 micrographs from the beta-galactosidase dataset [30] for a balanced dataset that was used to evaluate our approach.

### 3.2. Evaluation Metrics

In general, the signal-to-noise ratio is the way that the noise signal is measured in either a signal or an image. In other words, a better way to assess the amount of noise in an image is to measure the ratio of pure pixels (called mean of the signal or mean of the pixel values) to noisy pixels, which is the standard deviation of the signal (called the noise standard deviation), as shown in Equation (7).
(7)$$SNR=\frac{Mean\;signal}{Noise\;Standard\;Deviation}=\frac{{\left(\frac{1}{n}\sum_{i=1}^{n}x_{i}^{2}\right)}^{\frac{1}{2}}}{\sqrt{\frac{1}{n-1}}\sum_{i=1}^{n}{\left(x_{i}-{\bar{x}}_{i}\right)}^{2}},$$
where *n* is the total number of pixels in the micrograph.

In order to reduce radiation damage to the biomolecules of interest during the imaging process of the microscopy, a limited electron dose was used, as high-energy electrons can greatly damage specimens during imaging, resulting in extremely noisy micrographs. Moreover, the micrographs contained two-dimensional projections of a particle in different orientations. Generally, cryo-EM images have low contrast, due to the similarity of the electron density of the protein to that of the surrounding solution, as well as the limited electron dose used in data collection. In addition, the micrographs might have contained sections of ice, deformed particles, protein aggregates, etc., which would complicate particle picking. The cryo-EM images (micrographs) of the protein particles were taken by electron microscope, which contained randomly arranged particles along with non-particles—bits of frost, deformed particles, protein aggregates, and so on. These images suffered from heavy background noise and low contrast, due to a limited electron dose used in imaging. For these reasons, we observed that the micrographs had an extremely low signal-to-noise ratio.

Typically, for the first pre-processing stage, we used the EMAN2 software [29] to adjust the global intensity of the cryo-EM and convert them from MRC file format to the PNG image format in order to apply standard image-processing tools to them. In terms of selecting the best results, we used different scaling factors with EMAN2 [29] to adjust the intensity of the cryo-EM images. Then, we computed different evaluation metrics, such as the peak signal-to-noise ratio (PSNR), signal-to-noise ratio (SNR), and mean squared error (MSE), to evaluate the improvement of the quality of cryo-EM images in the whole dataset in each different scaling factor and compare the results with the original micrographs (i.e., without the using the intensity adjustment scaling factor).

Based on the SNR evaluation metric in Equation (7), where noise signal was measured, the PSNR often measured ratios between the maximum signal (pure pixels) and noise (corrupted pixels). PNSR uses a logarithmic decibel scale to measure the ratio between the maximum signal and noise that has a very wide dynamic range, as shown in Equation (8).
(8)$$SNR=10\times log_{10}\left(\frac{MAX_{i}^{2}}{MSE}\right),$$
where $MAX_{i}$ is the maximum possible pixel value of the micrograph and MSE is mean squared error given in Equation (9):(9)$$SNR=\frac{1}{m\times n}\sum_{i=1}^{n}\sum_{j=1}^{n}(\left|I\left(i,j\right)-\hat{I}\left(i,j\right)\right|),$$
where m×n is the micrograph image size, I is the original micrograph, and $\hat{I}$ is the pre-processed micrograph.

For the pre-processing stage, we used common image pre-processing criteria, such as peak signal-to-noise ratio (PSNR), signal-to-noise ratio (SNR), and mean squared error (MSE), to evaluate the improvement of the quality of cryo-EM images [33]. For the particle clustering and detection stages, we used the accuracy, precision, recall, and F1-score (i.e., the geometry means of precision and recall) to evaluate the particle clustering/detection results.

Table 1 reports the average quality measurements of the cryo-EM images with/without EMAN2 [29] intensity adjustment. The average quality measurements (PSNR, SNR, and MSE) of the original cryo-EM images were 28.06, 6.99 dB, and 26218.13, respectively. The intensity adjustment with many scaling factors improved the quality. The best scaling factor, which increased the PSNR and SNR while simultaneously decreasing the MSE, was the “sane” option, which picked a good range of scaling factors automatically. The three scores of using the “sane” scaling factor were improved by 29.27, 8.10 dB, and 0.198643.

Figure 12a compares the average PSNR and SNR scores of the cryo-EM images before and after all the pre-processing steps in the pre-processing stage. Figure 12b shows the MSE scores of the cryo-EM images before and after all the pre-processing steps.

The average PSNR score increased from 77.43 to 78.57 and the average SNR score increased from 3.40 to 4.05. The average MSE was reduced from 0.302 to 0.233. The range of PSNR scores increased from [77.429–77.43] for the original cryo-EM images to [78.52–78.64] for the pre-processed ones. The range of the SNR scores increased from [3.36–3.44] for the original images to [4.04–4.052] for the pre-processed ones. The range of MSE scores decreased from [0.3026–0.3033] to [0.23–0.237] after the pre-processing steps. According to Student’s t test, the p-values of the changes of PSNR, SNR, and MSE scores caused by the pre-processing were (-15.70) -28.31, and -19.53, respectively, indicating that the pre-processing steps significantly improved the quality of cryo-EM images.

### 3.3. Particle Clustering, Detection, and Picking Results

In order to evaluate the performance of automated particle clustering and picking, we generated a true reference by manually picking the particles on the images. Figure 13 illustrates the entire workflow of the super-clustering approach for fully automated complex and irregular single particle picking in cryo-EM.

The super-clustering approach was designed for fully automated single particle picking in cryo-EM. Our framework contained three stages. The first stage was micrograph pre-processing (shown on the yellow box of Figure 13), the second stage was the clustering stage, which had two different approaches, i.e., the regular clustering approach (shown in the blue box of Figure 13) and the super-clustering approach (shown in the orange box of Figure 13), and the third stage was the single particle picking (shown in the green box of Figure 13).

Figure 14 shows some examples of the fully automated complex single particle shape detection and picking using the super-clustering methods and the base cluster methods. The true particles that failed to be detected (false negatives) are denoted by red dots. Yellow dots represent the non-particle background (e.g., icy) objects that were falsely detected as particles (false positives).

Compared with the results of the base clustering methods (k-means, FCM, and IBC), the performances of the super-clustering methods (SP-IBC, SP-K-means, and SP-FCM) were significantly improved. The number of false negatives and false positives was significantly reduced. Table 2 reports the recall, precision, accuracy, F1 score, and running time of the base single particle picking methods (IBC, k-means, and FCM). Table 3 shows the results of the super-clustering methods (SP-IBC, SP-k-means, and SP-FCM).

Table 3 shows the results of the super-clustering methods (SP-IBC, SP-k-means, and SP-FCM). The three super-clustering methods were fully automated. The mean average of particle picking accuracy increased by 12.03%, 3.5%, and 2.1% for the IBC, k-means, and FCM, respectively. Also, the average time taken (pre-processing, running time of clustering, and particle picking) over the whole dataset decreased by 5.68 s for the SP-IBC, 86.24 s for the SP-k-means, and 59.58 s for the SP-FCM.

Generally, the super-clustering methods achieved better performance than their corresponding base methods according to almost all the metrics. SP-k-means clustering achieved a higher accuracy (95.48%) than SP-FCM (94.08%) and SP-IBC (88.98%). SP-IBC ran substantially faster than the other methods and all three super-clustering methods were fully automated.

### 3.4. Comparison with Other Particle Picking Methods

We compared SuperCryoEMPicker with two other methods, Scipion [34] and EMAN2 [29], in terms of computational efficiency, detection quality, and automation. Both Scipion [34] and EMAN2 [29] needed a reference set of particles to be selected manually (Figure 15a for Scipion; Figure 15e for EMAN2 [29]), which were used to train the methods to pick more particles (Figure 15d using Scipion and Figure 15e using EMAN2 [29]). Use of the arbitrarily manually selected particles resulted in most of the true particles being selected (Figure 15c using Scipion [34] and Figure 15e,f using EMAN2). However, some false positives, likely corresponding to thick ice, were also incorrectly selected (Figure 15d using Scipion [34] and Figure 15e using EMAN2 [29]). Increasing the number of the manually selected particles could reduce the number of false positives, but at the expense of increasing the number of false negatives (Figure 15e for EMAN2 [29]). In comparison, SuperCryoEMPicker successfully captured all true particles on the images without using any manually selected samples for training (Figure 15g–i).

Quantitative assessment of the comparison is shown in Figure 16 and Table 4. Figure 16a,b shows a micrograph from the beta-galactosidase dataset [30] after particle picking using EMAN2 [29], where only five particle references were selected (more references selected means more time taken to complete the task but more accurate results), and our super-clustering approach. Figure 16a shows the particle picking performance results using EMAN2 [29]. In terms of evaluating each particle picking tool, in addition to our fully automated particle picking approach, three criteria were selected to label and evaluate the particle picking performance results: True Positive (TP) picking, where the correct particles were marked by the yellow circles; False Negative (FN) picking, where the missed particles were marked by red circles; False Positive (FP) picking, where the incorrectly picked particles were marked by blue circles. Figure 16b shows the same criteria of the particle picking results using the super-clustering approach.

Table 4 illustrates the statistical evaluation of the performance results based on the TP, FN, and FP for each single particle picking algorithm, as well as the particle shape class and total number of particles (ground truth) in each image. Note that the super-clustering approach for fully automated single particle picking performed better in regard to detecting the shapes; it achieved 99.13% sensitivity, 98.45% precision, and 97.61% accuracy.

## 4. Conclusions

In this work, we designed SuperCryoEMPicker, a fully automated super-particle clustering method to pick particles of complex and irregular shapes in cryo-EM images. SuperCryoEMPicker, which was based on the super-clustering methods, was more accurate and ran faster than particle picking based on the base clustering methods. It also performed well compared to established semi-automated particle picking methods, which require users to manually pick reference particles for training. Therefore, SuperCyroEMPicker may be a useful and reliable tool for automated single particle picking in cryo-EM images.

## Figures and Tables

**Figure 1 genes-10-00666-f001:**
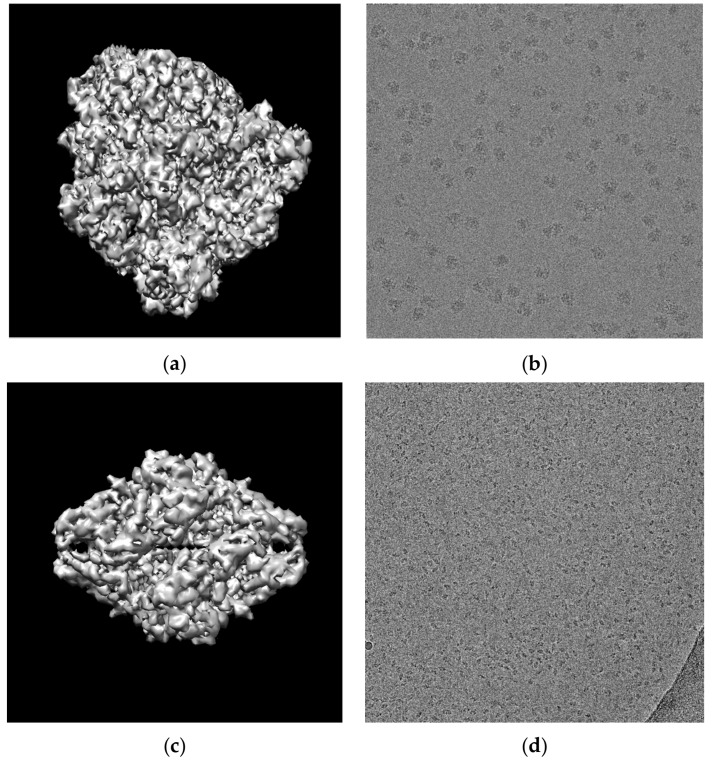
Examples of particles of complex and irregular shapes. (**a**) 80S ribosome [28] cryo-electron microscopy (cryo-EM) electron density map; (**b**) cryo-EM image of 80S ribosome particles (Electron Microscopy Public Image Archive (EMPIAR) entry 10028, [28]); (**c**) beta-galactosidase [30] cryo-EM electron density map; (**d**) cryo-EM image of beta-galactosidase particles (EMPIAR-10017, [30]). Both (**a**) and (**c**) were created using Chimera [31] and density maps from Protein Data Bank in Europe EMBL-EBI (The European Bioinformatics Institute) [32].

**Figure 2 genes-10-00666-f002:**
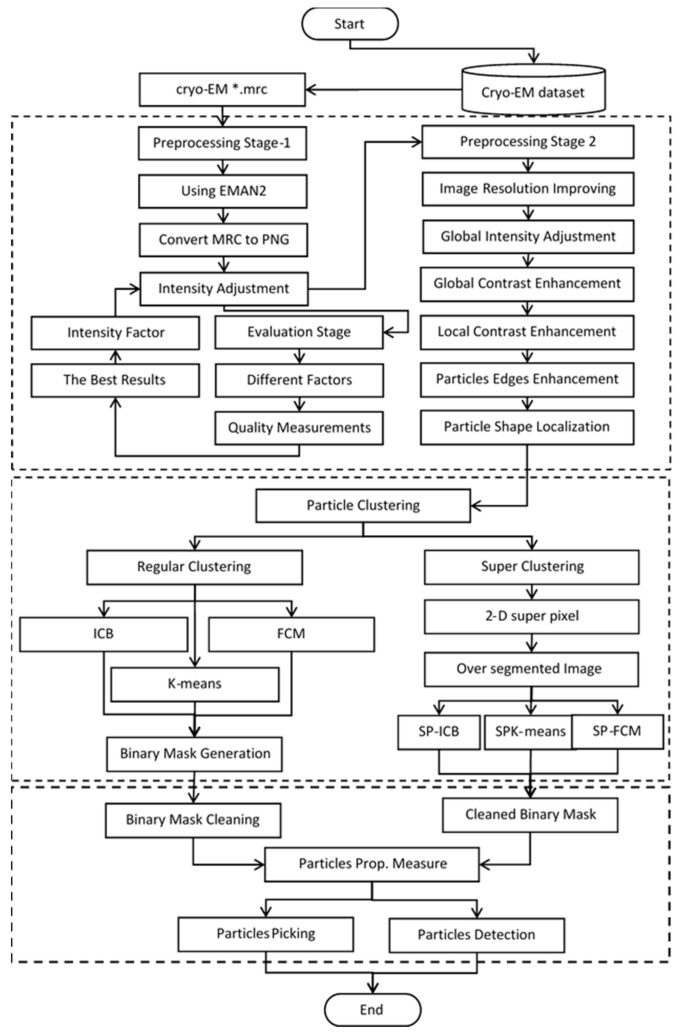
The general framework of the SuperCryoEMPicker. The dashed boxes represent three stages of the approach, i.e., pre-processing, super-clustering, and particle picking. Solid boxes denote analysis steps.

**Figure 3 genes-10-00666-f003:**
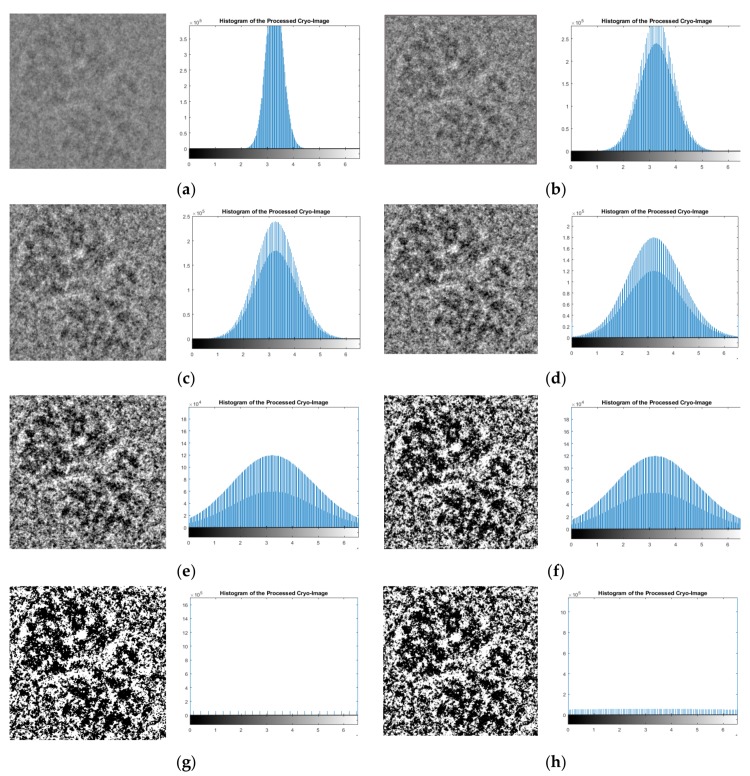
One zoomed-in particle image from the ribosome dataset [28] during the first stage of the pre-processing “intensity adjustment” using different scaling factors in the EMAN2 [29]. (**a**) Original zoomed-in particle (manually selected and cropped from the original image) and the original histogram of the image; (**b**) particle image after intensity adjustment using scale factor 5 and the histogram of the pre-processed image; (**c**) particle image after intensity adjustment with scale factor 4 and the histogram of the pre-processed image; (**d**) particle image after the intensity adjustment with scale factor 3 and the histogram of the pre-processed image; (**e**) particle after intensity adjustment with scale factor 1 and the histogram of the pre-processed image; (**f**) particle image after intensity adjustment with scale factor 0.1 and the histogram of the pre-processed image; (**g**) particle image after intensity adjustment with scale factor 0.25 and the histogram of the pre-processed; (**h**) particle image after intensity adjustment with scale factor 0.5 and the histogram of the pre-processed image.

**Figure 4 genes-10-00666-f004:**
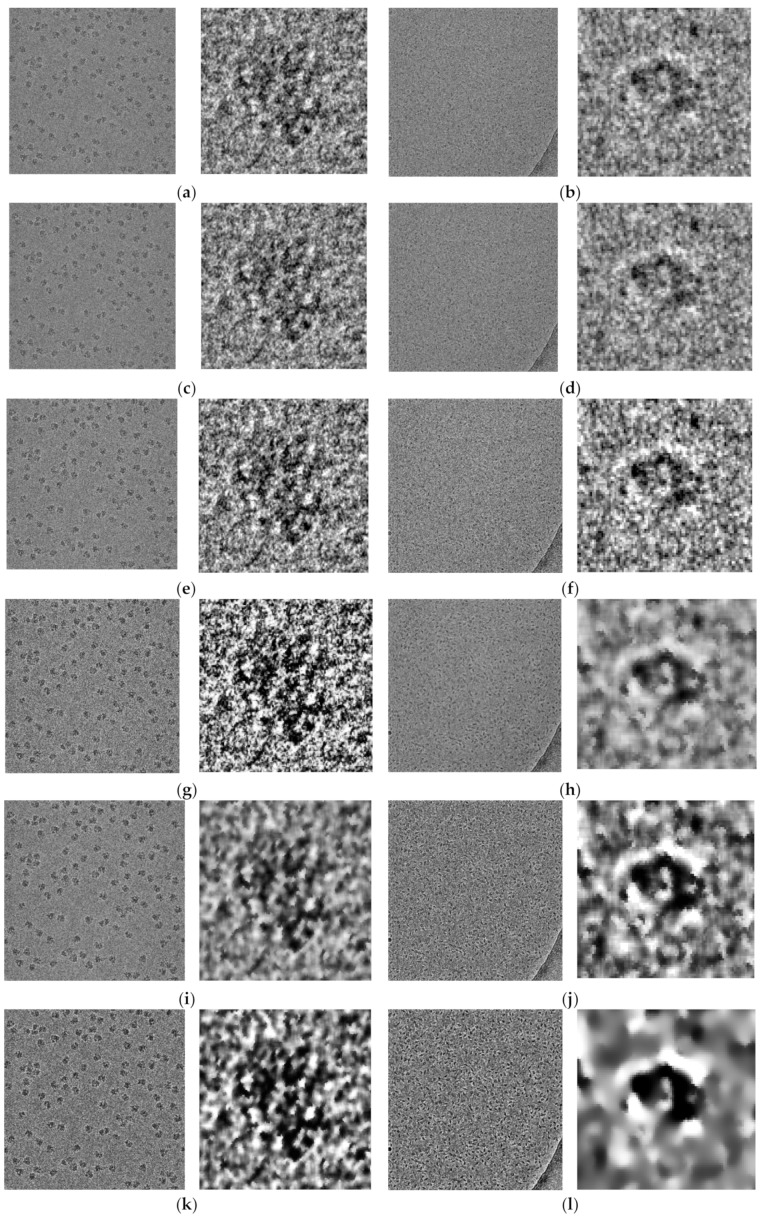
Illustration of the effects of the pre-processing procedures on ribosome [28] and beta-galactosidase [30] images. (**a**) Original particle image of ribosome [28] (one full image and one zoomed-in particle); (**b**) original image of beta-galactosidase [30]; (**c**) image of ribosome [28] after image resolution improvement; (**d**) image of beta-galactosidase [30] after image resolution improvement; (**e**) image of ribosome [28] after global intensity adjustment; (**f**) image of beta-galactosidase [30] after global intensity adjustment; (**g**) image of ribosome [28] after global contrast enhancement-based histogram equalization; (**h**) image of beta-galactosidase [30] after global contrast enhancement-based histogram equalization; (**i**) image of ribosome [28] after noise-suppression using the Wiener filter; (**j**) image of beta-galactosidase [30] after noise-suppression using the Wiener filter; (**k**) image of ribosome [28] after local particle contrast enhancement with adaptive histogram equalization; (**l**) image of beta-galactosidase [30] after local particle contrast enhancement using adaptive histogram equalization; (**m**) image of ribosome [28] after edge enhancement using guided image filtering; (**n**) image of beta-galactosidase [30] after edge enhancement using guided image filtering; (**o**) image of ribosome [28] after particle shape localization using morphological image operation; (**p**) image of beta-galactosidase [30] after particle shape localization using morphological image operation.

**Figure 5 genes-10-00666-f005:**
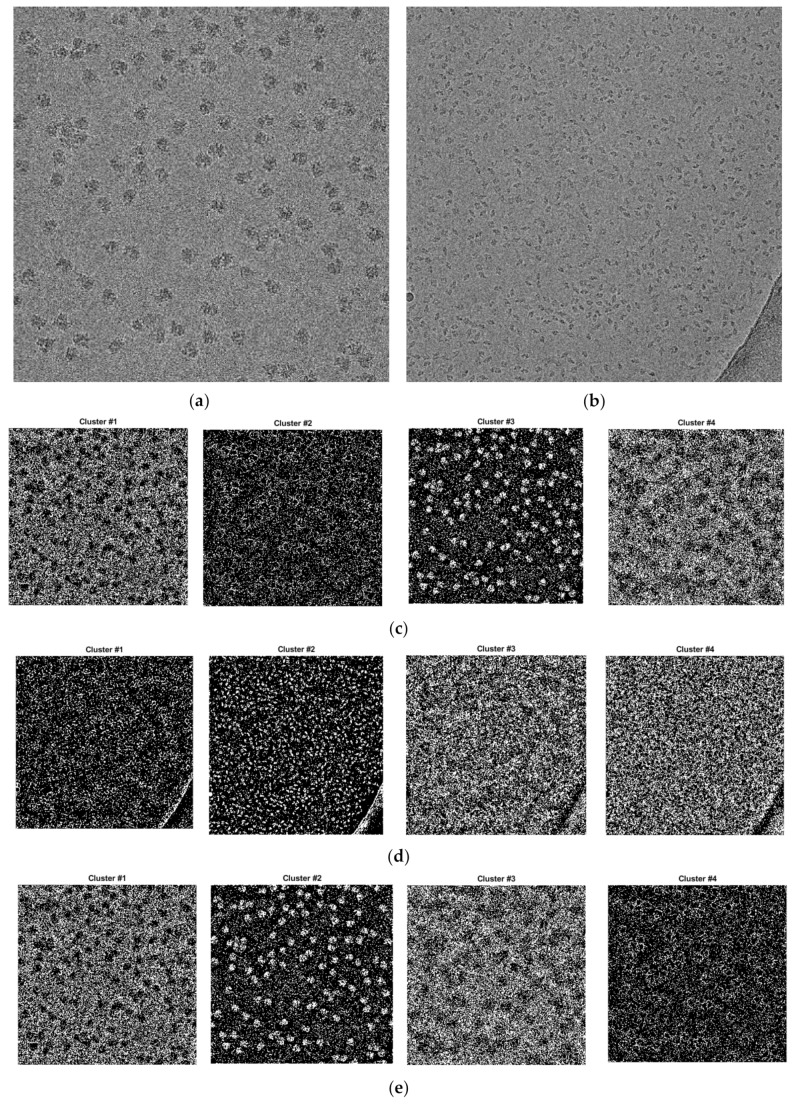
Image clustering results using k-means, FCM, and IBC. (**a**) Original cryo-EM image of the 80S ribosome [28]; (**b**) original cryo-EM image of beta-galactosidase [30]; (**c**) k-means clustering results (Cluster 1, Cluster 2, Cluster 3, and Cluster 4) for the ribosome [28] image; most real particles were assigned to Cluster 3; (**d**) k-means clustering results (Cluster 1, Cluster 2, Cluster 3, and Cluster 4) for the beta-galactosidase [30] image; most real particles were assigned to Cluster 2; (**e**) FCM ((Fuzzy C-means) clustering results (Cluster 1, Cluster 2, Cluster 3, and Cluster 4) for the ribosome [28] image; most real particles were assigned to Cluster 2; (**f**) FCM clustering results (Cluster 1, Cluster 2, Cluster 3, and Cluster 4) for the beta-galactosidase [30] image; most real particles were assigned to Cluster 1; (**g**) IBC (Intensity-Based Clustering) results (Cluster 1, Cluster 2, Cluster 3, and Cluster 4) for the ribosome [28] image. Most real particles were assigned to Cluster 1; (**h**) IBC clustering results (Cluster 1, Cluster 2, Cluster 3, and Cluster 4) for the beta-galactosidase [30] image. Most real particles were assigned to Cluster 1.

**Figure 6 genes-10-00666-f006:**
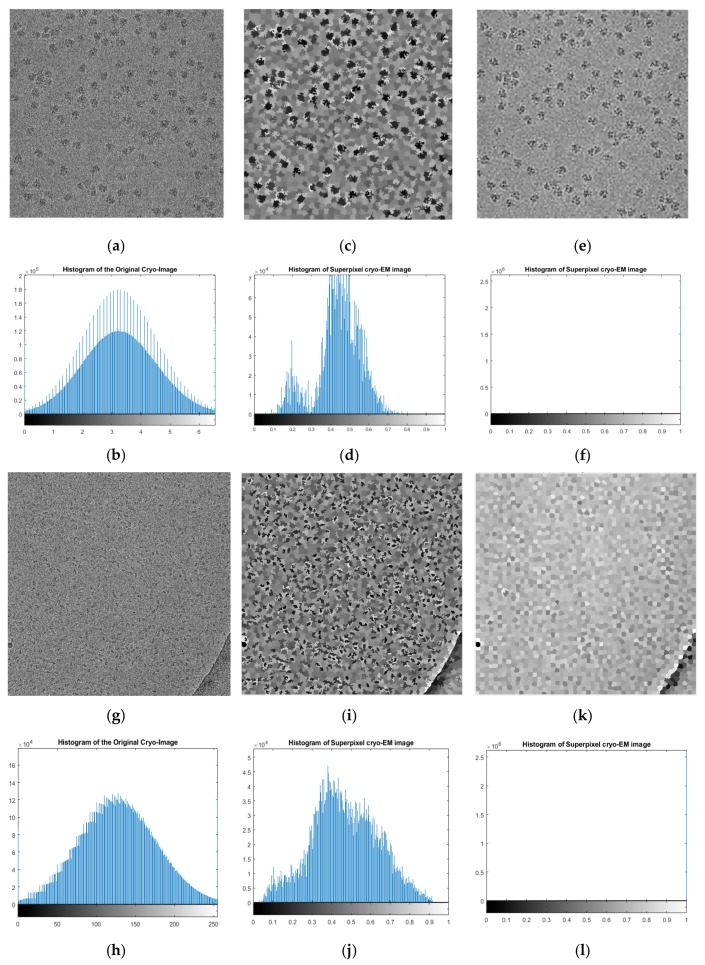
Different cryo-EM intermediate micrographs generated using simple linear iterative clustering (SLIC). (**a**) Original cryo-EM image of ribosome; (**b**) histogram of (**a**); (**c**) intermediate micrograph generated using simple linear iterative clustering (SLIC) based on the pre-processed image; (**d**) histogram of (**c**); (**e**) intermediate micrograph generated by SLIC from the original image; (**f**) histogram of (**e**); (**g**) original cryo-EM image of beta-galactosidase; (**h**) histogram of (**g**); (**i**) intermediate micrograph generated by SLIC from the pre-processed image; (**j**) histogram of (**i**); (**k**) intermediate micrograph generated by SLIC from the original image; (**l**) histogram of (**k**).

**Figure 7 genes-10-00666-f007:**
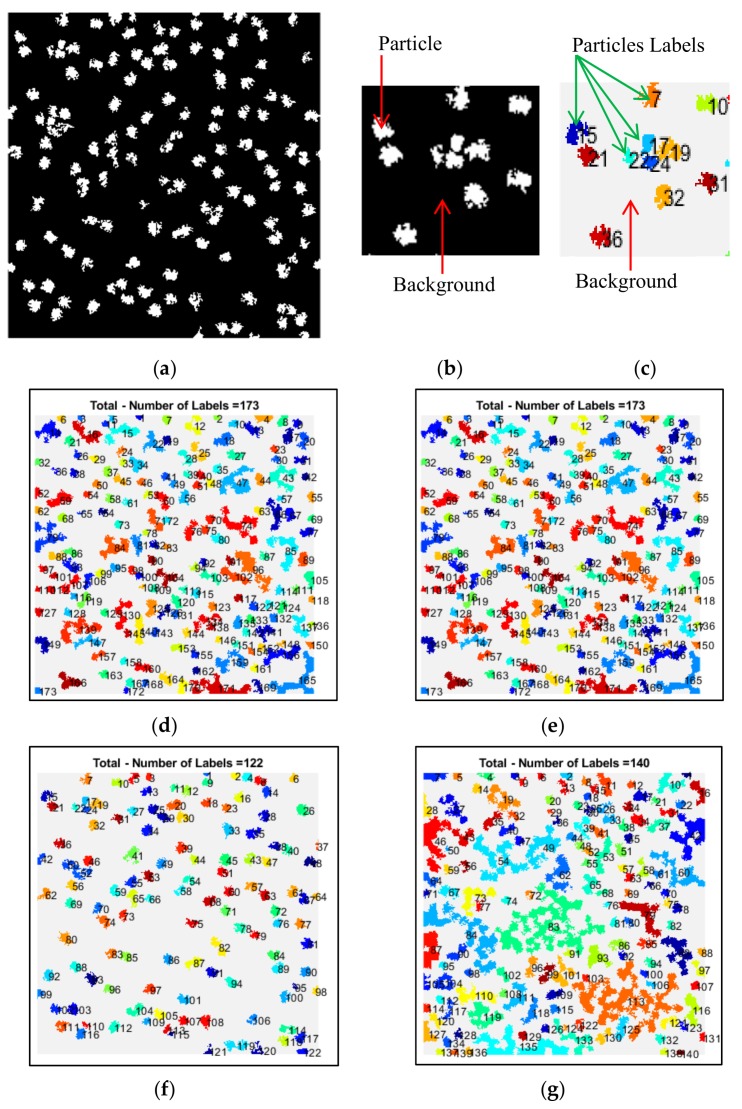
SP-k-means evaluation and automated cluster section based on extracting the total number of the particles in each cluster and selecting the cluster that has the minimum total number of particles. (**a**) Binary clustered image; (**b**) Particle labels in the clustered image; (**c**) Label of each individual particle number in the clustering evaluation stage for the fully automated cluster image selection; (**d**) Total number of objects (particles) in the Cluster index 1; (**e**) total number of objects (particles) in Cluster index 2; (**f**) total number of objects (particles) in Cluster index 3; (**g**) total number of objects (particles) in Cluster index 4. The super-pixel k-means clustering (SP-k-means) is shown in Algorithm 1.

**Figure 8 genes-10-00666-f008:**
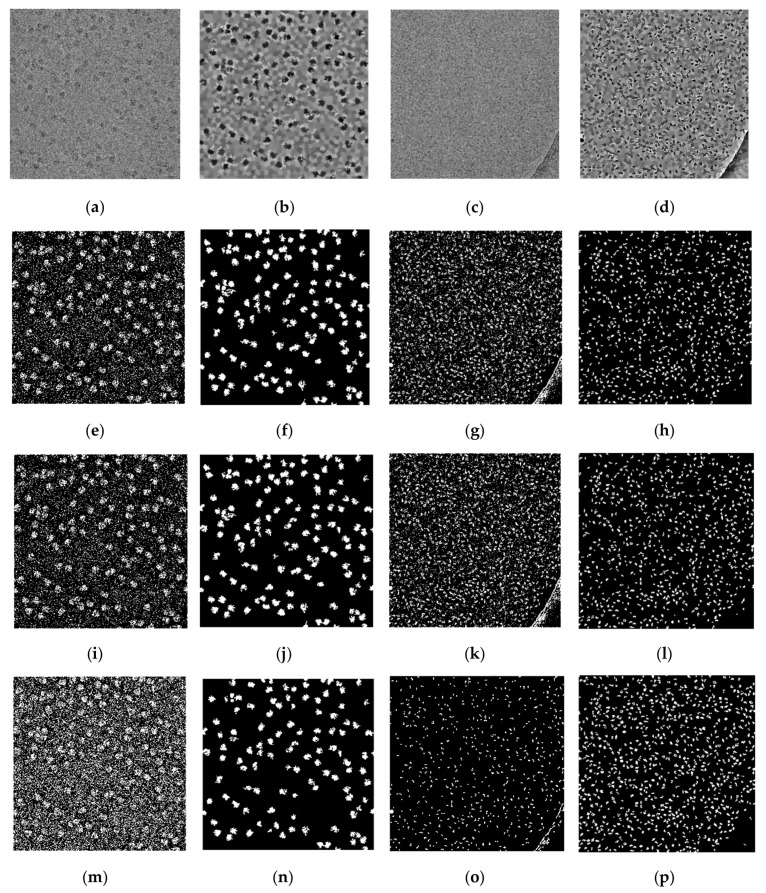
Cryo-EM super-clustering results of SP-k-means, SP-FCM, and SP-IBC in comparison with the base algorithms. (**a**) An original cryo-EM image of the ribosome [28]; (**b**) intermediate micrograph generated by SLIC from pre-processed image of the ribosome [28]; (**c**) original cryo-EM image of beta-galactosidase [30]; (**d**) intermediate micrograph generated by SLIC for beta-galactosidase [30]; (**e**) k-means clustering results of the ribosome [28]; (**f**) SP-k-means clustering results of the ribosome [28]; (**g**) k-means clustering results of beta-galactosidase [30] cryo-EM image; (**h**) SP-K-means clustering results of beta-galactosidase [30] cryo-EM image; (**i**) FCM clustering results of the ribosome [28] cryo-EM image; (**j**) SP-FCM clustering results of the ribosome cryo-EM image; (**k**) FCM clustering results of the beta-galactosidase cryo-EM image; (**l**) SP-FCM clustering results of the beta-galactosidase [30] cryo-EM image; (**m**) IBC clustering results of the ribosome [28] cryo-EM image; (**n**) SP-IBC clustering results of the ribosome [28] cryo-EM image; (**o**) IBC clustering results of the beta-galactosidase [30] cryo-EM image; (**p**) SP-IBC clustering results of the beta-galactosidase [30] cryo-EM image.

**Figure 9 genes-10-00666-f009:**
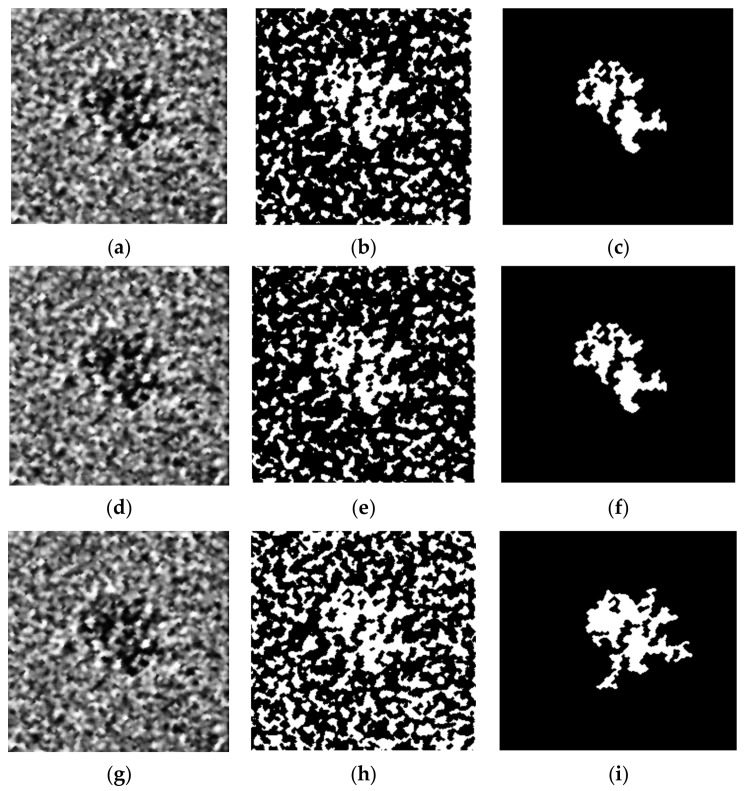
A zoomed-in selected particle image before and after binary image cleaning and non-connected object removal on the image masks generated by the three base clustering methods. (**a**) Original zoomed-in particle image; (**b**) particle clustering image before binary image cleaning by k-means for the ribosome [28]; (**c**) particle clustering image after binary image cleaning by k-means for the ribosome [28]; (**d**) original zoomed-in particle image; (**e**) particle clustering image before binary image cleaning by FCM clustering for the ribosome [28]; (**f**) particle clustering image after binary image cleaning by FCM clustering for the ribosome [28]; (**g**) original zoomed-in particle image; (**h**) particle clustering image before binary image cleaning by IBC clustering for the ribosome [28]; (**i**) particle clustering image after binary image cleaning by IBC clustering for the ribosome [28].

**Figure 10 genes-10-00666-f010:**
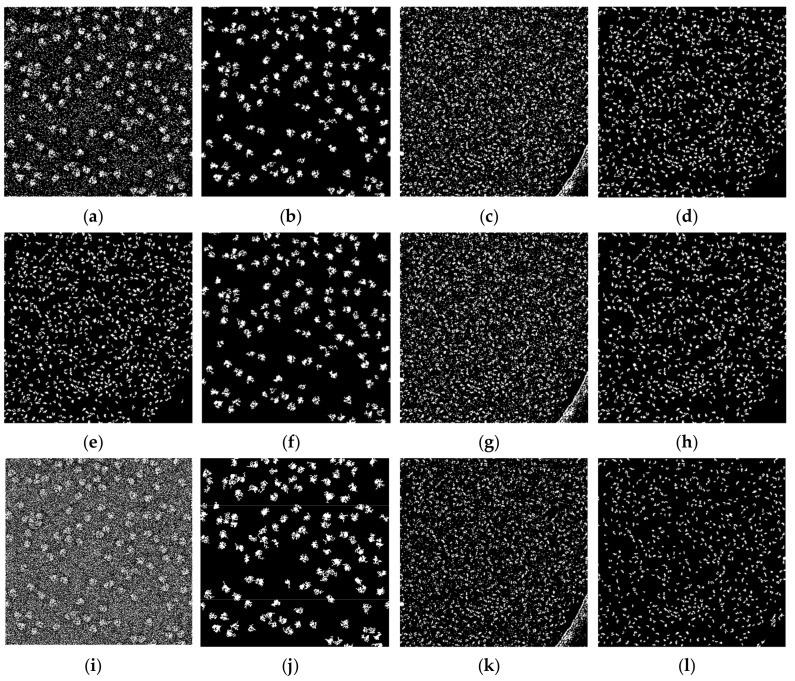
Whole cryo-EM particle clustering results before and after binary image cleaning for the three base clustering methods. (**a**) Ribosome [28] clustered image of k-means; (**b**) image mask of (**a**) after image cleaning; (**c**) beta-galactosidase [30] clustered image of k-means; (**d**) binary image mask of (**c**) after image cleaning; (**e**) ribosome [28] clustered image of FCM; (**f**) binary image mask of (**e**) after image cleaning; (**g**) beta-galactosidase [30] clustered image of FCM; (**h**) binary mask of (**g**) after image cleaning; (**i**) ribosome [28] clustered image of IBC; (**j**) binary image mask of (**i**) after image cleaning; (**k**) beta-galactosidase [30] clustered image of IBC; (**l**) binary image mask of (**k**) after image cleaning.

**Figure 11 genes-10-00666-f011:**
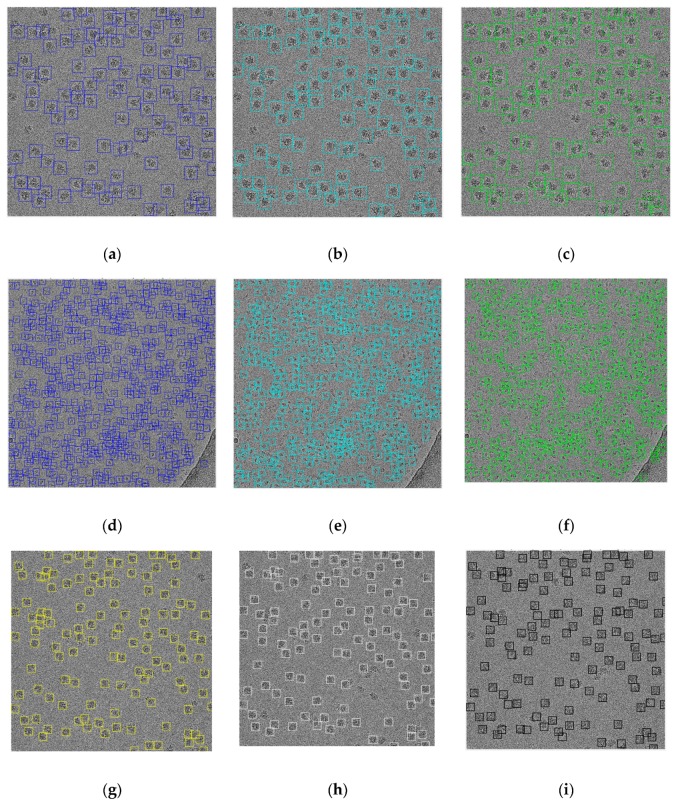
Results of detecting particles of irregular and complex shapes on ribosome [28] and beta-galactosidase [30] images. (**a**) Particle detection and picking by k-means on the ribosome [28] image; (**b**) particle detection and picking by FCM on the ribosome [28] image; (**c**) particle detection and picking by IBC on the ribosome [28] image; (**d**) particle detection and picking by k-means on the beta-galactosidase [30] image; (**e**) particle detection and picking by FCM on the beta-galactosidase [30] image; (**f**) particle detection and picking by IBC on the beta-galactosidase [30] image; (**g**) particle detection and picking by SP-k-means on the ribosome [28] image; (**h**) particle detection and picking by SP-FCM on the ribosome [28] image; (**i**) particle detection and picking by SP-IBC on the ribosome [28] image; (**j**) particle detection and picking by SP-K-means on the beta-galactosidase [30] image; (**k**) particle detection and picking by SP-FCM on the beta-galactosidase [30] image; (**l**) particle detection and picking by SP-IBC on the beta-galactosidase [30] image.

**Figure 12 genes-10-00666-f012:**
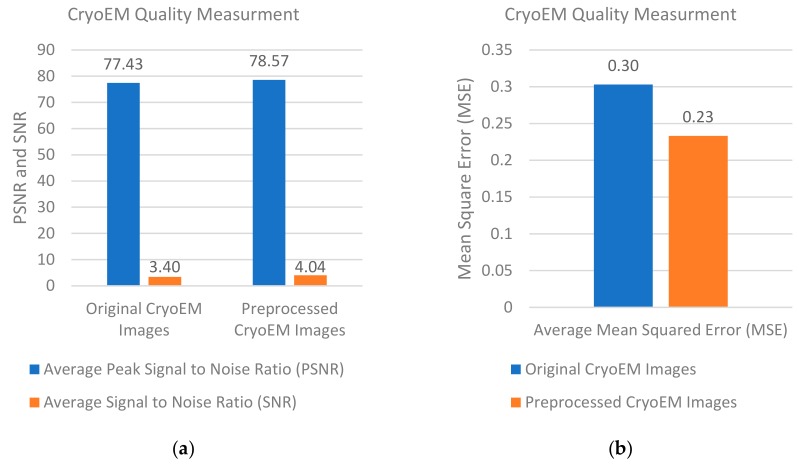
Quality of cryo-EM images before and after the pre-processing stage. (**a**) Average PSNR and SNR values of the cryo-EM images before and after the pre-processing steps; (**b**) average MSE values of the cryo-EM images before and after the pre-processing steps.

**Figure 13 genes-10-00666-f013:**
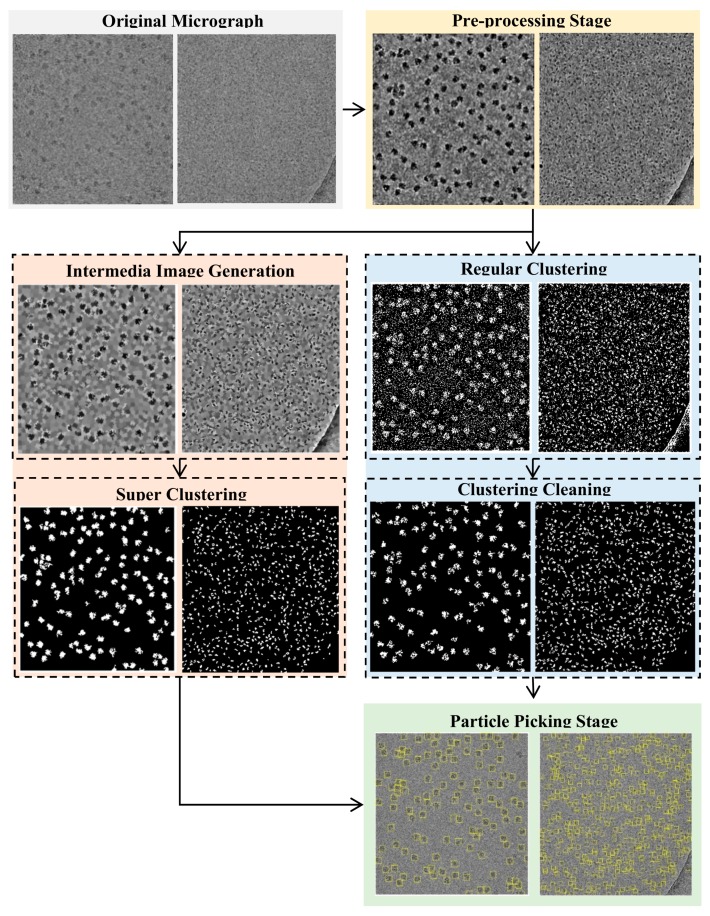
Entire workflow of the super-clustering approach for fully automated complex and irregular single particle picking in cryo-EM.

**Figure 14 genes-10-00666-f014:**
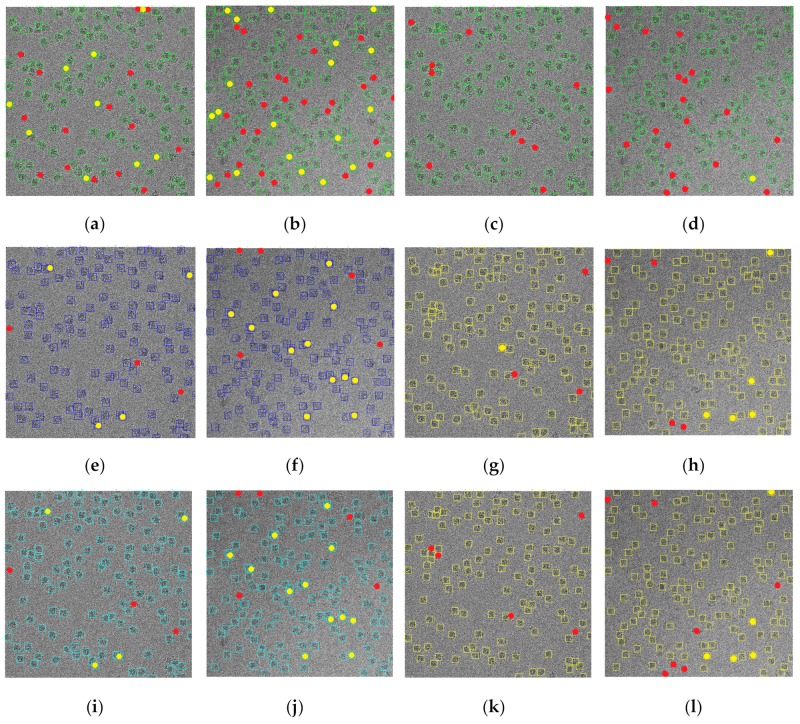
Results of the fully automated single particle picking in cryo-EM images by the base and super-clustering methods. Red dots denote the missing particles not detected (false negatives) and yellow dots show false positives. (**a**) Particle picking of IBC; (**b**) particle picking of IBC on an extremely low SNR cryo-EM image; (**c**) particle picking of SP-IBC; (**d**) single particle picking of SP-IBC on an extremely low SNR cryo-EM image; (**e**) particle picking of k-means; (**f**) particle picking of k-means clustering algorithm on an extremely low-SNR cryo-EM image; (**g**) particle picking of SP-k-means; (**h**) particle picking of SP-k-means on an extremely low-SNR cryo-EM image; (**i**) particle picking of FCM; (**j**) particle picking of FCM on an extremely low-SNR cryo-EM image; (**k**) particle picking of SP-FCM; (**l**) particle picking of SP-FCM on an extremely low-SNR cryo-EM image.

**Figure 15 genes-10-00666-f015:**
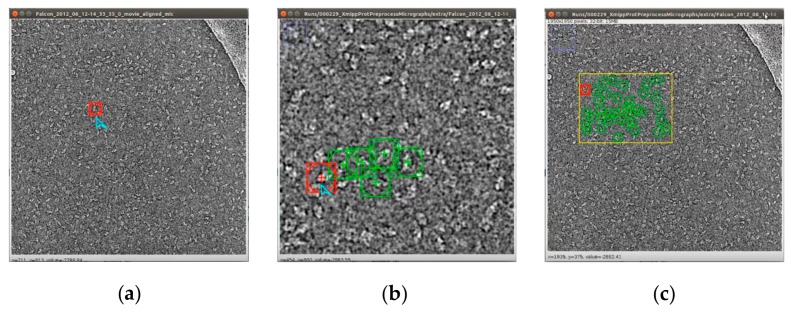
Particle picking using EMAN2 [29], Scipion [34], and SuperCryoEMPicker. (**a**) Manually selected reference particle of the beta-galactosidase [30] image for Scipion [34]; (**b**) zoom-in view of some manually selected reference particles for the beta-galactosidase [30] image for Scipion [34]; (**c**) final reference particles of beta-galactosidase manually selected for Scipion [34]; (**d**) all particle picking results of Scipion [34] trained on 40 manual reference particles on the image of the beta-galactosidase; (**e**) EMAN2 [29] autopicking results based on different manually selected training samples in the first tested image of the beta-galactosidase dataset [30]; (**f**) manually selected reference particles of beta-galactosidase [30] for EMAN2 [29]; (**g**) particle picking results of SuperCryoEMPicker based on SP-IBC clustering; (**h**) particle picking results of SuperCryoEMPicker based on SP-k-means clustering; (**i**) particle picking results of SuperCryoEMPicker based on SP-FCM clustering.

**Figure 16 genes-10-00666-f016:**
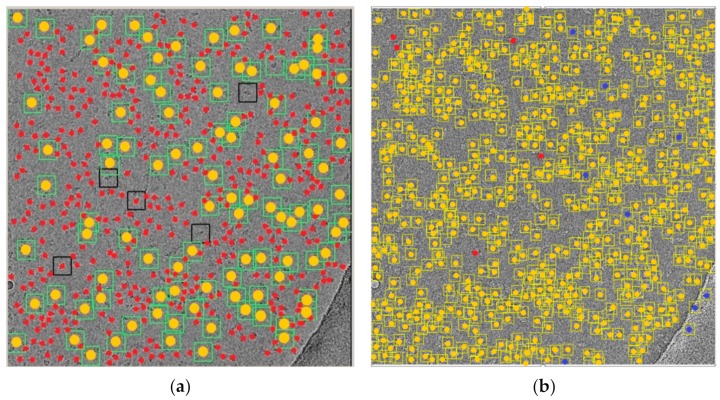
Evaluation of particle picking using EMAN2 and the super-clustering approach for fully automated single particle picking. (**a**) Particle picking results of the beta-galactosidase image [30] using EMAN2 [29]; (**b**) particle picking results of the beta-galactosidase image [30] using the super-clustering approach for fully automated single particle picking. Particles are labeled as follows: Yellow—True Positive (TP); red—False Negative (FN); blue—False.

**Table 1 genes-10-00666-t001:** The average peak signal-to-noise ratio (PSNR), signal-to-noise ratio (SNR), and mean squared error (MSE) of the cryo-EM images with or without EMAN2 [29] intensity adjustment, according to different scaling factors.

Intensity Adjustment Scaling Factor	PSNR	SNR
Original Image (without Adjustment)	28.06118	6.99260 dB
Scaling Factor = 10	27.55033	20.00023 dB
Scaling Factor = 5	28.06118	13.98521 dB
Scaling Factor = 4	28.34026	12.05305 dB
Scaling Factor = 3	28.84182	9.59165 dB
Scaling Factor = 2	29.91141	6.43044 dB
Scaling Factor = 1	32.61282	2.87674 dB
Scaling Factor = 0.1	44.37584	0.16536 dB
Scaling Factor = 0.5	35.81839	1.26105 dB
Scaling Factor = 0.25	39.23404	0.55222 dB
Scaling Factor = Sane	29.26647	8.09887 dB

**Table 2 genes-10-00666-t002:** Results of particle picking using the base clustering algorithms (k-means, FCM, and IBC).

Measures	IBC	k-Means	FCM
Sensitivity/Recall (%)	82.42	97.14	97.14
Precision (%)	87.07	94.50	94.50
Accuracy (%)	76.95	91.98	91.98
F1 Score (%)	84.68	95.80	95.80
Time consuming (s)	16.77	113.55	412.86
Automation	Fully Automated	Manual	Manual

**Table 3 genes-10-00666-t003:** Results of the super-clustering methods (SP-k-means, SP-FCM, and SP-IBC).

Measures	IBC	k-Means	FCM
Sensitivity/Recall (%)	92.44	97.50	95.71
Precision (%)	96.62	97.86	98.19
Accuracy (%)	88.98	95.48	94.08
F1 Score (%)	94.48	97.68	96.93
Time Taken (s)	11.09	27.31	353.28
Automation	Fully Automated	Fully Automated	Fully Automated

**Table 4 genes-10-00666-t004:** Statistical evaluation super-clustering approach for fully automated single particle picking and EMAN2 [29] performances using a beta-galactosidase image [30]. The table reports True Positive (TP) picking results where the correct particles are picked, False Negative (FN) picking results where some good particles are missed, and False Positive (FP) picking results where the incorrect particles (other objects such as background or artificial objects) are picked as particles.

Evaluation Metric	Our Approach	EMAN2
Total Particles Number	579	579
True Positive (TP)	574	101
False Negative (FN)	5	478
False Positive (FP)	9	0
Sensitivity/Recall (%)	0.99136442	0.174439
Precision (%)	0.98456261	1
Accuracy (%)	0.97619048	0.174439
F1 Score (%)	0.98795181	0.297059

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
