# Peer review of "A Super-Clustering Approach for Fully Automated Single Particle Picking in Cryo-EM"

_genes, 2019, doi:10.3390/genes10090666_

Round 1
Reviewer 1 Report
The topic of the initial clustering/identification of objects in cryo-EM imgaes is timely. The super clustering algorithm the authors present improves the quality of cryo-EM images compared to the standard techniques, which select the first patters manually. The presented method can therefore speed up the initial particle picking and relieve a bottleneck in the cryo-EM structure determination. The improved pipeline is demonstrated using two examples, ribosome particles, and beta-galactosidase. The examples should be better described, what datasets are being analyzed, where the data were retrieved, in what format. References 29 and 30 appear late in the text, should be also cited in figures.
Figure 12: poorly formatted numbers in histograms.
References are presented in different formats.
Author Response
Reviewer 1
Comments:
The topic of the initial clustering/identification of objects in cryo-EM imgaes is timely. The super clustering algorithm the authors present improves the quality of cryo-EM images compared to the standard techniques, which select the first patters manually. The presented method can therefore speed up the initial particle picking and relieve a bottleneck in the cryo-EM structure determination. The improved pipeline is demonstrated using two examples, ribosome particles, and beta-galactosidase. The examples should be better described, what datasets are being analyzed, where the data were retrieved, in what format. References 29 and 30 appear late in the text, should be also cited in figures.
Authors Response:Thank you for the great comments. In the revised version, references 29 and 30 have been cited in figures.
Comments:
Figure 12: poorly formatted numbers in histograms.
Authors Response:Thank you for the great comments. In the revised version, the histogram numbers in Figure 12 have been reformatted.
Comments:
References are presented in different formats.
Authors Response:Thank you for the great comments. In the revised version, all the references have been reformatted.
Reviewer 2 Report
See attached file.

Reviewer 3 Report
General Consideration:
This paper frames some very interesting mathematical methods for detection and picking of particles in cryo-EM images. Well done to the authors.
My only general concern is that algorithm performance is not mentioned outside of tables 2 and 3, and the numbers therein are not elaborated on. Are these timings per-image or per-dataset? Timing is important to consider in data processing in cryo-EM, as less accurate heuristic methods can be more valuable than more thorough ones just through time and compute cost savings alone.
Minor Considerations:
Data Input
In the manuscript, the authors refer to inputs images being 3D. Typically I would operate a particle picker on flattened 2D micrographs, not raw frame stacks. Does this picker act on frames and flattened images? If so, is there a benefit to picking on the entire frame stack?
I noticed that the code base is in Matlab, and that you are using external program (EMAN) to convert to PNG for processing. Have you attempted to use Matlab extensions for loading MRC files directly to save time? (E.G. : https://www.mathworks.com/matlabcentral/fileexchange/27021-imagic-mrc-dm-and-star-file-i-o)
Prose
Paragraph at line 221 - I believe intermedia should be intermediate in this paragraph
Figures and Tables
Figure 2 - I would recommend grouping the labels of images and histograms together, it might read easier.
Figure 10 - panel (e) is mentioned twice in the legend and appears to be out of place (the image is a beta-gal, not a ribosome image)
Table 3 - Column headers are pasted from the previous table and not updated to reflect what is shown
Author Response
Comments:
My only general concern is that algorithm performance is not mentioned outside of tables 2 and 3, and the numbers therein are not elaborated on. Are these timings per-image or per-dataset? Timing is important to consider in data processing in cryo-EM, as less accurate heuristic methods can be more valuable than more thorough ones just through time and compute cost savings alone.
Authors Response:Thank you for the great comments. The timings (runtimes) are the average of the entire datasets. More details have been added in the revised version specifically in the second paragraph of section 3.3 “Particle Clustering, Detection and Picking Results” in page 22.
Comments:
In the manuscript, the authors refer to inputs images being 3D. Typically I would operate a particle picker on flattened 2D micrographs, not raw frame stacks. Does this picker act on frames and flattened images? If so, is there a benefit to picking on the entire frame stack?
I noticed that the code base is in Matlab, and that you are using external program (EMAN) to convert to PNG for processing. Have you attempted to use Matlab extensions for loading MRC files directly to save time? (E.G. : https://www.mathworks.com/matlabcentral/fileexchange/27021-imagic-mrc-dm-and-star-file-i-o)
Authors Response:Thank you for the great comments. Based on the whole framework that we have proposed. The first preprocessing step is done by using the EMAN2 which takes the 3D raw frame stacks and converted to 2D flattened micrographs. Then, the second part of our framework (designed) has performed the particle picking on the 2D flattened micrographs which generally speaking by combining the two parts together it does the particle picking of the average frames (2D micrograph). The main benefit of doing the picking on the entire frame stack (from 3D to 2D) that we don’t need to do the Contrast Transfer function (CTF) and correct each frame which is a critical and time consuming step to increase the micrograph resolution by cryo-electron microscopy (cryoEM) reconstruction. In our case, we propose such less time-consuming steps to increase the resolution and enhance the quality of the micrographs. More details about the 2D pre-processing can be founded in our first model “AutoCryoPicker” on the first stage “Stage 1: pre-processing, Step 1: Cryo-EM image resolution improving” [1].
Adil Al-Azzawi, Anes Ouadou, John J. Tanner, Jianlin Cheng, AutoCryoPicker: an unsupervised learning approach for fully automated single particle picking in Cryo-EM images, BMC Bioinformaticsvolume 20, Article number: 326 (2019). https://bmcbioinformatics.biomedcentral.com/articles/10.1186/s12859-019-2926-y
Comments:
I noticed that the code base is in Matlab, and that you are using external program (EMAN) to convert to PNG for processing. Have you attempted to use Matlab extensions for loading MRC files directly to save time? (E.G. : https://www.mathworks.com/matlabcentral/fileexchange/27021-imagic-mrc-dm-and-star-file-i-o)
Authors Response:Thank you for the great comments. We have tried to use the MATLAB directly to read MRC but we got a shifted data result. In future, we will try to solve the shifting data problem by reading the MRC to our system directly without the EMAN2, The provided link is useful thanks for that.
Comments:
Paragraph at line 221 - I believe intermedia should be intermediate in this paragraph
Authors Response:Thank you for the great comments. It has been fixed in page 221.
Comments:
Figure 2 - I would recommend grouping the labels of images and histograms together, it might read easier.
Authors Response:Thank you for the great comments. The figures and histograms have been grouped together and new description of Figure 3 has been added in the revised version between line 121-132 in page 5.
Comments:
Figure 10 - panel (e) is mentioned twice in the legend and appears to be out of place (the image is a beta-gal, not a ribosome image)
Authors Response:Thank you for the great comments. The image of the ribosome has been fixed in Figure 10, page 18 in the revised version.
Comments:
Table 3 - Column headers are pasted from the previous table and not updated to reflect what is shown
Authors Response:Thank you for the great comments. More details for Table 3 have been added in the revised version between line 408-414 in page 23.
Round 2
Reviewer 2 Report
The authors have mostly addressed my concerns. I recommend publication in Genes.